# Research on the Power Loss of High-Speed and High-Load Ball Bearing for Cryogenic Turbopump

**Wenhu Zhang [1], Chaojie Zhang [1,\*], Xusheng Miao [2], Liang Li [3] and Sier Deng [1]**

[1]  School of Mechatronics Engineering, Henan University of Science and Technology, Luoyang 471000, China
[2]  Xi'an Aerospace Propulsion Institute, Xi'an 710012, China
[3]  Luoyang Bearing Research Institute Corporation Limited, China National Machinery Industry Corporation, Luoyang 471039, China
**\***  Correspondence: zhangchaojst@163.com

**Abstract:** This paper studies the lubrication characteristics of ball bearings for cryogenic turbopumps. First, the frictional coefficients between 440C and a Ag coating, 440C and solid PTFE (polytetrafluoroethylene), and 440C and a PTFE coating in LN2 (liquid nitrogen) are obtained using a ball-on-disk testing machine under a high sliding speed in the range of 0 to 8 m/s and a high contact stress in the range of 2.5 to 3.5 GPa. Dynamic and power loss models of high-speed and high-load ball bearings are established to study the key factors affecting the heat generation characteristics. In order to verify the correctness of these two theoretical models, a coupled fluid-thermal finite element model is built to evaluate the temperatures of the outer ring under different bearing speeds, which are then proved by experiments with ball bearings for cryogenic turbopumps. The results show that the power loss due to the spinning-sliding of the ball and the churning and drag of LN2 account for more than 80% of the total power loss; the spin-roll ratio of the ball on the inner raceway is a key indicator for this type of ball bearing, and the relatively small radial clearance and contact angle are suggested. Both of the proposed theoretical models have sufficient accuracy and can be used in the performance evaluation and optimization design of bearings.

**Keywords:** cryogenic turbopump; ball bearing; dynamic model; power loss; coupled fluid-thermal finite element model



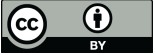

## 1. Introduction

Liquid propellant rockets with high performance are a key element of orbital launch systems [1,2]. To reduce launch costs and increase efficiency, advanced rocket engines are required. Among the many factors, ball bearings are the key component to support the rotation of the cryogenic turbopump within the rocket engine [3,4]. Analysis of actual systems shows that most breakdowns of the cryogenic turbopump are connected with the failure of the ball bearings, which means that the performance of bearings directly determines the lifespan of the orbital launch system.

Gibson H G [5] reviewed the history of space shuttle main engine turbopump bearing testing at the Marshall Space Flight Center, describing the typical failure characteristics of bearings and analyzing the failure mechanism through a metallurgical analysis. In order to meet the expected life goals of bearings in space shuttle main engine high-pressure oxidizer turbopumps, Gibson H G [6] developed a new bearing material with a better manufacturing technique, improvement of cage materials for better lubrication, and substitution of the steel ball with a silicon nitride ball to improve the wear resistance and thermal stability of the bearing. Moore L [7] tested hybrid ceramic rolling element bearings in liquid hydrogen at Marshall Space Flight Center to evaluate their long-term durability for use at high-speed in cryogenic rocket engine turbomachinery. Dufrane K F [8] measured the film thickness between two contact surfaces of bearings in an LN2 environment and demonstrated that the contact surfaces are not effectively separated by the film. Nosaka, Oike, Kikuchi [9–11]

studied the self-lubricating properties, film-forming, and film-breaking properties of the PTFE transfer film of high-speed ball bearings in a LE-7 rocket engine. Subsequently, Nosaka [12–14] studied the wear, transfer film, and self-lubricating performance of hybrid ceramic ball bearings for rocket turbopumps, and analyzed the effects of an iron fluoride layer on the durability of the high-speed ball bearings. Chang et al. [15] carried out the experiments with a Hertzian pressure in the range of 2.0 to 3.0 GPa and with a high rolling velocity of up to 48 m/s and analyzed the effects of materials and surface roughness on the scuffing characteristics of rolling/sliding contacts cooled and lubricated with LO2 (liquid oxygen). Wang Liqin [16] tested the frictional characteristics of self-lubricating hybrid ceramic ball bearings in LN2. The results showed that pitting was found on the steel races, and the power loss of a hybrid ceramic ball bearing is lower than that of an all-steel ball bearing. Nosaka [17] tested a hybrid ceramic bearing consisting of Si3N4 balls and an all-steel bearing in LH2 (liquid hydrogen) at speeds of up to 120,000 rpm, 3 million DN (bearing bore diameter/mm × speed/rpm), under thrust loads of up to 3140 N and sealed pressures of up to 1.7 MPa. The authors studied the effect of cooling medium flow on the temperature and evaluated the critical load capacity for the hybrid ceramic bearings and all-steel bearings. Deng Sier, Ma Meiling [18] discussed the key factors affecting ultra-low temperature bearing in a liquid rocket engine and carried out structural optimization design of the bearings using the multi-objective optimization method. Li Hongliang, Zhang Xu [19] analyzed the reasons for ring bands and blackish charring appearing on the surface of a steel ball after tests and pointed out that the flash temperature and instant oxidation are the primary causes. Masataka [20] presented a topical review of previous cryogenic tribology studies on the research and development of bearings for LH2 turbopumps and analyzed the friction, wear characteristics, and existing friction problems of low-temperature ball bearings in LE-5 and LE-7. Servais C [21] performed tests on a cryotechnic ball bearing in a LO2 environment, and found that a sudden increase in temperature occurred beyond a critical loading of the bearing; the reasons for which were explained. V Vartha [22] made a failure analysis of the ball bearings of a turbopump in a liquid rocket engine and presented the root cause of the bearing failure and recommendations to avoid such failures. Choe B [23] used computational fluid dynamics software to calculate the fluid force of the cage and studied the influence of the ball-cage pocket and guide clearances on the trajectory of the center of mass of the cage under different drag coefficients. The same group [24,25] used experimental methods to study the effects of guidance clearance, cage ball-pocket clearance, and dynamic imbalance of ball bearings on the core trajectory, the vortex frequency of the cage, and the frictional moment and wear state of the bearing under an ultra-low temperature environment. Wang Q [26] used a ball-on-disk tribometer to study the effects of temperature, load, and sliding speed on the tribological properties of PTFE composites in cryogenic temperatures and demonstrated that the friction coefficients fall as the load increases. Based on a test with a ball bearing with a PTFE retainer and a Ag coating on the surfaces of the raceways, Miao [27] analyzed the PTFE transfer film and the wear of the Ag coating and revealed the formation and the evolution of the PTFE transfer film. Kwak W [28] proposed a hydrodynamic friction model considering the hydrodynamic effect of cryogenic fluids via a ball-on-disk experiment under the Hertzian pressure of 1.114 GPa and with a slip velocity in the range of 0.2 to 3.0 m/s, and experimentally studied the dynamic behavior of a cryogenic bearing. Through experiments, Su Bing [29] studied the frictional characteristics of 440C-Ag coating, 440C-PTFE coating, and 440C-solid PTFE under ultra-low temperatures, high sliding speed (8 m/s), and high contact stress (3.5 GPa). Xia Z [30] tested a fully ceramic ball bearing under cryogenic conditions and heavy loads and analyzed the tribological behavior of the ball bearing. Su H [31] presented an improved quasi-static model for cryogenic bearings in a liquid hydrogen pump considering the effect of centrifugal expansion and shrinkage on the bearing rings and studied the effect of the rotating speed and axial preload on dynamic characteristics and friction of bearing. Gupta and Gibson [32,33] established a dynamic analysis model of cryogenic ball bearings based on the experimental study of the 440C drag coefficient [34], and studied the heat generation

of the ball bearing, which were validated by experiments, and made a comparison of total power loss between the all-steel ball bearing and a hybrid ceramic ball bearing.

To sum up, most research into ball bearings for cryogenic turbopump incline towards experimental studies, with theoretical studies being relatively scarce, especially for high-speed and high-load ball bearing. In light of this, this paper focuses on a high-speed and high-load ball bearing for use in cryogenic turbopumps. Firstly, the frictional coefficients of the bearing's contact surfaces are measured using ball-on-disk experiments. Dynamic, power loss and coupled fluid-thermal finite element models for the ball bearing in the cryogenic turbopump are established to study the key factors governing heat generation and evaluate the temperature of the outer ring, the results of which are verified by experiment with ball bearings in a cryogenic turbopump. This study will contribute to the performance evaluation and optimization design of high-speed and high-load ball bearings for cryogenic turbopumps.

## 2. Theoretical Models of Ball Bearing for Cryogenic Turbopump

### 2.1. Frictional Coefficients of Bearing's Contact Interfaces

For ball bearings in cryogenic turbopumps, a PTFE cage is applied to provide lubrication in the cryogenic liquid. Due to the fact that the PTFE transfer film originating from the PTFE cage cannot immediately form at the beginning of their operation, it is often desirable to deposit a solid lubricant coating with a certain thickness on the raceways to provide bearing lubrication. When the bearing runs for a while, the PTFE is transferred to the balls through contacts between the balls and the PTFE cage and is further transferred to the raceways continuously through contacts between the balls and the inner/outer raceway. The lubrication states of the contact surfaces in the different working phases of a ball bearing are shown in Figure 1.

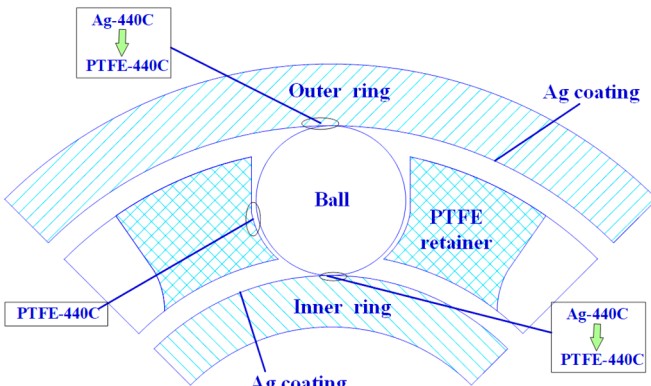

**Figure 1.** The lubrication states of bearing contact surfaces in the different working stages.

In this experiment, the experimental time is 200 s, the ambient temperate is $-175\,^{\circ}$C, the sliding velocity ranges from 0 to 8.0 m/s, and the contact stress ranges from 2.5 to 3.5 GPa, both of which ranges are much larger than those in Ref. [28].

By using a ball-on-disc testing machine [29], the frictional coefficients of 440C-Ag coating, 440C-solid PTFE, and 440C-PTFE coating under different contact stresses and sliding speeds were measured. By fitting the experiment data, the frictional coefficient μ is expressed as a function of the sliding velocity in Equation (1), which is originally suggested by Kragelskii [35]. This equation is well-known and was widely used in many papers. At here, we draw the Figure 2 to show the variation tendency of the Equation (1) clearly.

$$\mu = (A + Bv)e^{-Cv} + D \qquad (1)$$

where $A$, $B$, $C$, and $D$ can be obtained by the sliding velocity $v$ and contact stress $P$, $K_m$ is the maximum frictional coefficient, $K_\infty$ is the frictional coefficient at infinite sliding velocity, and $S_m$ is the sliding velocity at the maximum frictional coefficient.

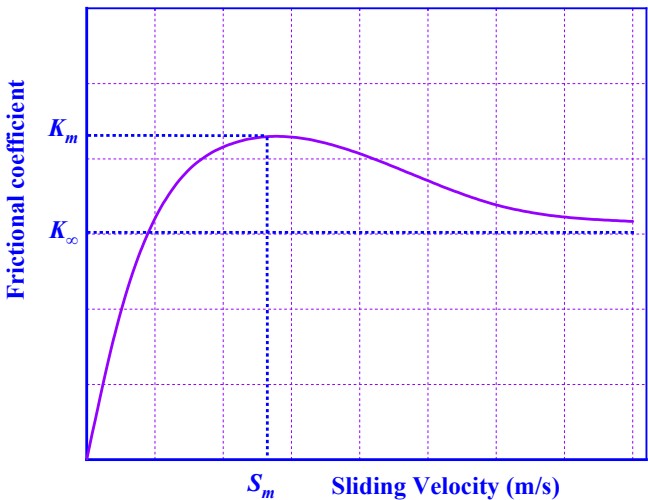

**Figure 2.** Change of frictional coefficient with sliding velocity [35].

The ball-disc test specimens are shown in Figure 3. The parameter values under different contact interfaces are shown in Table 1.

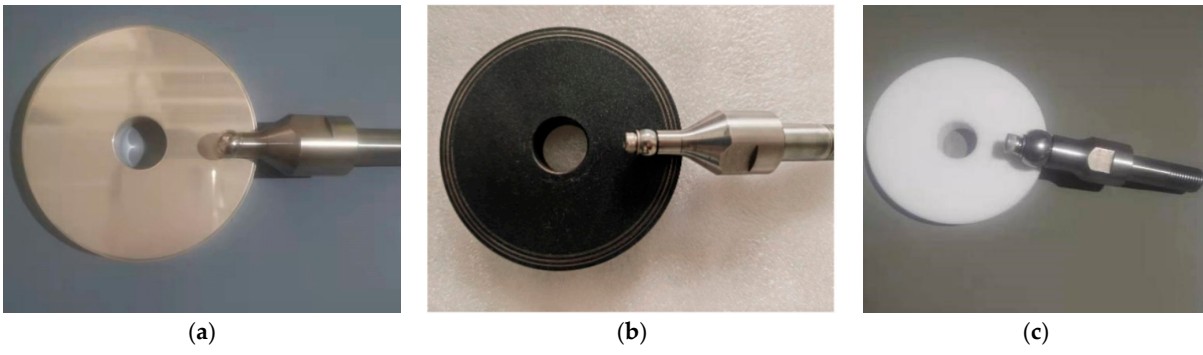

(**a**)  (**b**)  (**c**)

**Figure 3.** Ball-disc test specimens. (**a**) 440C-Ag-coated disk. (**b**) 440C-PTFE-coated disk. (**c**) 440C-solid PTFE disk.

**Table 1.** The values of $S_m$, $K_m$, $K_\infty$ under different contact interfaces.

| Contact Interface | 440C-Ag Coating | | | 440C-PTFE Coating | | | 440C-PTFE | | |
|---|---|---|---|---|---|---|---|---|---|
| Contact Stress | 2.5 GPa | 3.0 GPa | 3.5 GPa | 2.5 GPa | 3 GPa | 3.5 GPa | 12 MPa | 16 MPa | 20 MPa |
| $S_m$ (m/s) | 3.0 | 3.3 | 3.6 | 2.44 | 2.56 | 2.66 | 2.58 | 2.52 | 2.46 |
| $K_m$ | 0.138 | 0.128 | 0.121 | 0.130 | 0.126 | 0.119 | 0.125 | 0.127 | 0.137 |
| $K_\infty$ | 0.124 | 0.116 | 0.102 | 0.109 | 0.105 | 0.100 | 0.106 | 0.110 | 0.117 |

The comparisons between the computation results of Equation (1) and the experimental data for the 440C-Ag coating, 440C-PTFE coating, and 440C-solid PTFE are shown in Figure 4. The maximum relative errors are all less than 10%.

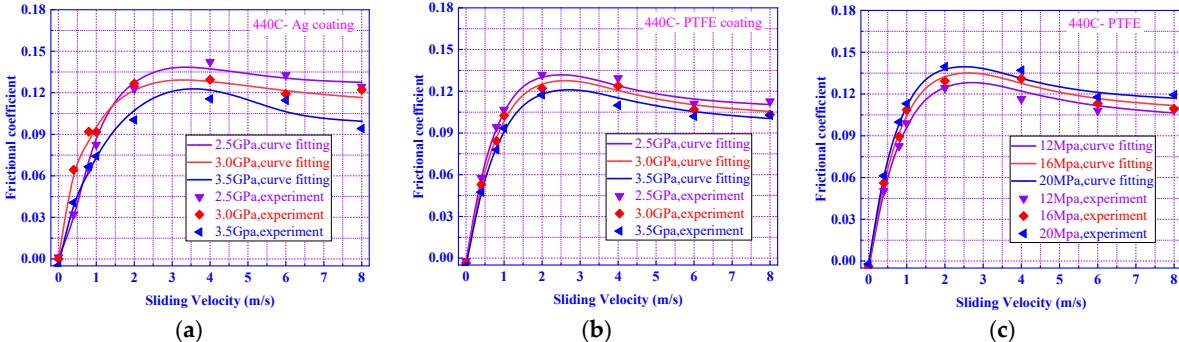

**Figure 4.** The comparisons between the computation results and the experimental data. (**a**) 440C-Ag coating. (**b**) 440C-PTFE coating. (**c**) 440C-solid PTFE.

### 2.2. Dynamic Model of Ball Bearing

### 2.2.1. Coordinate System

In this paper, the outer ring is fixed, the inner ring rotates, and the cage is guided by the outer ring. The component's mass center coincides with its centroid. In order to build the dynamic model of the ball bearings, five coordinate systems are defined, as shown in Figure 5 and explained in Table 2. In the figure and the table, the subscript $i$ represents the inner ring, $o$ represents the outer ring, $b$ represents the ball, $c$ represents the cage and $j$ represents the $j$th ball.

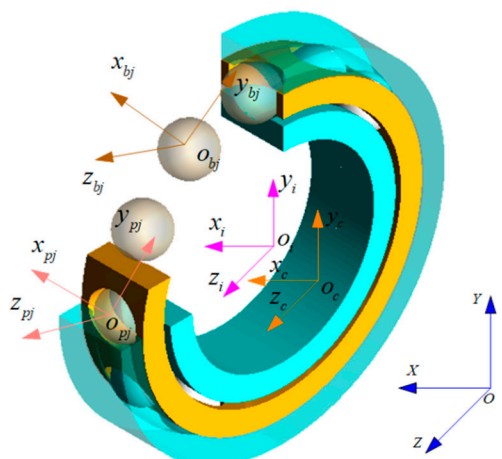

**Figure 5.** Coordinate systems of the ball bearings.

**Table 2.** Definitions of the coordinate systems.

| Coordinate System Name | Coordinate System Symbol | Coordinate System Definition |
| --- | --- | --- |
| Inertial coordinate system | $S_O = \{O; X, Y, Z\}$ | $X$-axis coincides with rotating axis of bearing, and $YZ$-plane parallels to radial plane through bearing center. |
| Coordinate system of the $j$th ball | $s_{bj} = \{o_{bj}; x_{bj}, y_{bj}, z_{bj}\}$ | $o_{bj}$ coincides with ball's mass center, $y_{bj}$ axis is along radial direction of bearing, and $z_{bj}$ axis is along circumferential direction of bearing. |
| Coordinate system of cage's | $s_c = \{o_c; x_c, y_c, z_c\}$ | $x_c$-axis coincides with rotating axis of cage, $y_c z_c$-plane parallels to radial plane through cage center, $o_c$ coincides with geometric center of cage. |
| Coordinate system of inner ring | $s_i = \{o_i; x_i, y_i, z_i\}$ | $x_i$-axis is along with rotating axis of inner ring, $y_i z_i$-plane parallels with radial plane through inner ring mass center, $o_i$ coincides with geometric center of inner ring. |
| Coordinate system of the $j$th cage pocket center | $s_{pj} = \{o_{pj}; x_{pj}, y_{pj}, z_{pj}\}$ | $o_{pj}$ coincides with geometric center of cage pocket, $y_{pj}$-axis is along radial direction of bearing, and $z_{pj}$-axis is along circumferential direction of bearing. |

### 2.2.2. Nonlinear Dynamics Differential Equations of the *j*th Ball

In Figure 6, $\eta$, $\xi$ denote the long axis and short axis of the contact zone, respectively; $\alpha$ is the contact angle between the ball and raceway; $\omega_b$ is the angular velocity of the ball, which can be decomposed into the spin angular velocity of the ball $\omega_{bx}$, $\omega_{by}$, $\omega_{bz}$; and $\omega_{so}$, $\omega_{si}$ are the spin angular velocities of the ball, respectively.

$$\omega_{so} = \omega_{by}\cos\alpha_o + \omega_{bz}\sin\alpha_o \tag{2}$$

$$\omega_{si} = \omega_{by}\cos\alpha_i + \omega_{bz}\sin\alpha_i \tag{3}$$

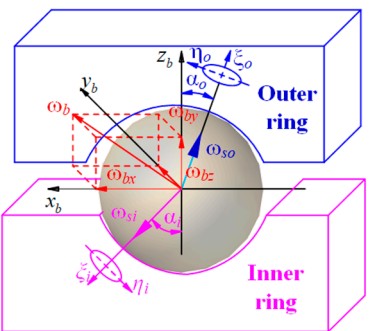

**Figure 6.** The motion of the ball.

The spin-roll ratio of the ball can be written as:

$$SR_o = \omega_{so}/\omega_{ro} \tag{4}$$

$$SR_i = \omega_{si}/\omega_{ri} \tag{5}$$

where $SR_o$ is the spin-roll ratio of the ball on the outer raceway; $SR_i$ is the spin-roll ratio of the ball on the inner raceway; and $\omega_{ro}$, $\omega_{ri}$ are the rolling angular velocities of the ball on the outer ring and inner ring, respectively.

When ball bearings are working at high speed, the forces acting on the *j*th ball are shown in Figures 7 and 8.

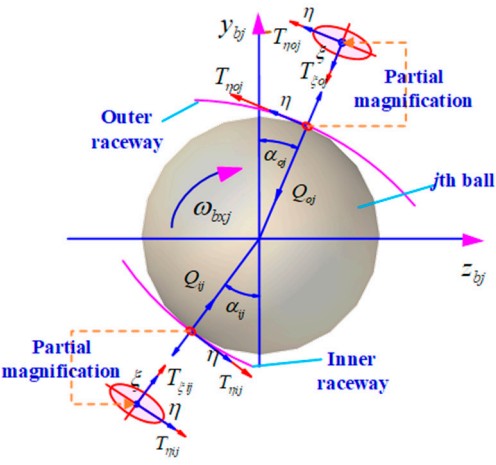

**Figure 7.** Forces between the *j*th ball and the raceways.

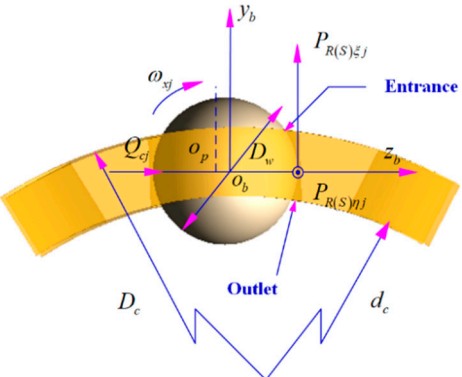

**Figure 8.** Forces between the $j$th ball and the cage.

Where $Q_{ij}$, $Q_{oj}$ are the normal contact forces between the $j$th ball and the raceways; $T_{\eta ij}$, $T_{\eta oj}$, $T_{\xi ij}$, $T_{\xi oj}$ are the frictional forces along the long axis and short axis of the contact zone between the $j$th ball and the raceways; $Q_{cj}$ is the collision force between the $j$th ball and the cage; $D_c$ is the outer diameter of the cage; $d_c$ is the inner diameter of the cage; and $D_p$ is the diameter of the cage pocket.

The forces acting on the cage are shown in Figure 9. $\omega_c$ is the orbital angular velocity; $e$ is the relative eccentricity of the cage center, $\Delta Yc$, $\Delta Zc$ are components of $e$ along the $Y$, $Z$ directions. $\phi_c$ is the angle between $y_c$ and $-Y$, $h_0$ is the minimum fluid film thickness, $F_{cx}$, $F_{cy}$ are components of the hydrodynamic force acting on the cage's surface along the $y_c$, $z_c$ directions; and $M_{cx}$ is the friction moment acting on the cage's surface.

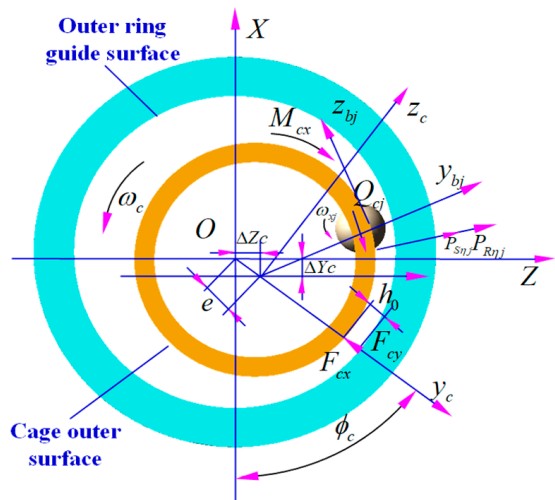

**Figure 9.** Forces acting on the cage.

The nonlinear dynamics differential equations of the $j$th ball are shown as Equations (6)–(11):

$$Q_{ij}\sin\alpha_{ij} - Q_{oj}\sin\alpha_{oj} + T_{\eta ij}\cos\alpha_{ij} - T_{\eta oj}\cos\alpha_{oj} + P_{S\xi j} + P_{R\xi j} = m_b\ddot{x}_{bj} \tag{6}$$

$$Q_{ij}\cos\alpha_{ij} - Q_{oj}\cos\alpha_{oj} - T_{\eta ij}\sin\alpha_{ij} + T_{\eta oj}\sin\alpha_{oj} + F_{\eta j} - P_{S\eta j} - P_{R\eta j} + F_{Nj}\sin\varphi_{bj} = m_b\ddot{y}_{bj} \tag{7}$$

$$T_{\xi oj} - T_{\xi ij} + Q_{cj} - F_{Dj} - F_{\tau j} + F_{Nj}\cos\varphi_{bj} = m_b\ddot{z}_b \tag{8}$$

$$T_{\xi oj}\cos\alpha_{oj}D_w/2 + T_{\xi ij}\cos\alpha_{ij}D_w/2 - \left(P_{S\eta j} + P_{R\eta j}\right)D_w/2 - J_x\dot{\omega}_{xj} = I_b\dot{\omega}_{bjx} \tag{9}$$

$$-T_{\xi oj}\sin\alpha_{oj}D_w/2 - T_{\xi ij}\sin\alpha_{ij}D_w/2 - G_{yj} - \left(P_{S\xi j} + P_{R\xi j}\right)D_w - J_y\dot{\omega}_{yj} = I_b\dot{\omega}_{bjy} - I_b\omega_{bjz}\dot{\theta}_{bj} \tag{10}$$

$$T_{\eta oj}D_w + T_{\eta ij}D_w/2 - G_{zj} - J_z\dot{\omega}_{zj} = I_b\dot{\omega}_{bjz} + I_b\omega_{bjy}\dot{\theta}_{bj} \tag{11}$$

where $m_b$ is the mass of the ball; $F_{Nj}$ is the centrifugal force of the $j$th ball; $\ddot{x}_{bj}$, $\ddot{y}_{bj}$, $\ddot{z}_{bj}$ are the displacement accelerations of the $j$th ball; $I_b$ is the moment of inertia of the ball; $\varphi_j$ is the azimuth angle of the $j$th ball; $\dot{\omega}_{xj}$, $\dot{\omega}_{yj}$, $\dot{\omega}_{zj}$ are components of the ball's angular acceleration in the $x_{bj}$, $y_{bj}$, $z_{bj}$ directions; $\omega_{bjx}$, $\omega_{bjy}$, $\omega_{bjz}$ are the angular velocities of the $j$th ball; $\dot{\omega}_{bjx}$, $\dot{\omega}_{bjy}$, $\dot{\omega}_{bjz}$ are the angular accelerations of the $j$th ball; $\dot{\theta}_{bj}$ is the orbit speed of the $j$th ball; $D_w$ is the ball diameter; $F_{\eta j}$, $F_{\tau j}$ are components of the ball's inertial force; $P_{R\eta j}$, $P_{R\xi j}$ are the rolling frictional forces acting on the ball's surface; $P_{S\eta j}$, $P_{S\xi j}$ are the sliding frictional forces acting on the ball's surface; $J_x$, $J_y$, $J_z$ are components of the ball's moment of inertia in the $x_{bj}$, $y_{bj}$, $z_{bj}$ directions; $G_{yj}$, $G_{yj}$ are components of the ball's moment of inertia in the $y_{bj}$, $z_{bj}$ directions; and $F_{Dj}$ is the resistance acting on the ball by the fluid medium.

### 2.2.3. Nonlinear Dynamics Differential Equations of the Cage

The nonlinear dynamics differential equations of the cage are shown as Equations (12)–(17):

$$\sum_{j=1}^{BN} \left( P_{S\eta j} + P_{R\eta j} + Q_{cxj} \right) = m_c \ddot{x}_c \tag{12}$$

$$\sum_{j=1}^{BN} \left[ \left( P_{S\xi j} + P_{R\xi j} \right) \cos \varphi_j + Q_{cyj} \right] + F_{cy} = m_c \ddot{y}_c \tag{13}$$

$$\sum_{j=1}^{BN} \left[ \left( P_{S\xi j} + P_{R\xi j} \right) \sin \varphi_j - Q_{czj} \right] + F_{cz} = m_c \ddot{z}_c \tag{14}$$

$$\sum_{j=1}^{BN} \left[ \left( P_{S\xi j} + P_{R\xi j} \right) \frac{D_w}{2} - Q_{cj} \frac{D_{pw}}{2} \right] + M_{cx} = I_{cx} \dot{\omega}_{cx} - \left( I_{cy} - I_{cz} \right) \omega_{cy} \omega_{cz} \tag{15}$$

$$\sum_{j=1}^{BN} \left[ \left( P_{S\eta j} + P_{R\eta j} \right) \frac{D_{pw}}{2} \sin \varphi_j \right] = I_{cy} \dot{\omega}_{cy} - \left( I_{cz} - I_{cx} \right) \omega_{cz} \omega_{cx} \tag{16}$$

$$\sum_{j=1}^{BN} \left[ \left( P_{S\eta j} + P_{R\eta j} \right) \frac{D_{pw}}{2} \cos \varphi_j \right] = I_{cz} \dot{\omega}_{cz} - \left( I_{cx} - I_{cy} \right) \omega_{cx} \omega_{cy} \tag{17}$$

where $m_c$ is the mass of the cage; $\ddot{x}_c$, $\ddot{y}_c$, $\ddot{z}_c$ are the displacement accelerations of the cage mass center; $I_{cx}$, $I_{cy}$, $I_{cz}$ are the moments of inertia of the cage; $\omega_{cx}$, $\omega_{cy}$, $\omega_{cz}$ are the angular velocities of the cage; $\dot{\omega}_{cx}$, $\dot{\omega}_{cy}$, $\dot{\omega}_{cz}$ are the angular accelerations of the cage; $Q_{cxj}$, $Q_{cyj}$, $Q_{czj}$ are components of $Q_c$; $BN$ is the ball number; and $D_{pw}$ is the pitch diameter of the bearing.

### 2.2.4. Nonlinear Dynamics Differential Equations of the Inner Ring

The nonlinear dynamics differential equations of the inner ring are shown in Equations (18)–(22):

$$F_x + \sum_{j=1}^{BN} Q_{ij} \sin \alpha_{ij} = m_i \ddot{x}_i \tag{18}$$

$$F_y + \sum_{j=1}^{BN} \left( Q_{ij} \cos \alpha_{ij} \cos \varphi_j + T_{\xi ij} \sin \varphi_j \right) = m_i \ddot{y}_i \tag{19}$$

$$F_z - \sum_{j=1}^{BN} \left( Q_{ij} \cos \alpha_{ij} \sin \varphi_j + T_{\xi ij} \cos \varphi_j \right) = m_i \ddot{z}_i \tag{20}$$

$$M_y + \sum_{j=1}^{BN} \left( r_{ij} Q_{ij} \sin \alpha_{ij} \sin \varphi_j + \frac{D_w}{2} f_i T_{\xi ij} \sin \alpha_{ij} \cos \varphi_j \right) = I_{iy} \dot{\omega}_{iy} - \left( I_{iz} - I_{ix} \right) \omega_{iz} \omega_{ix} \tag{21}$$

$$M_z + \sum_{j=1}^{BN} \left( r_{ij} Q_{ij} \sin\alpha_{ij} \cos\varphi_j - \frac{D_w}{2} f_i T_{\xi ij} \sin\alpha_{ij} \sin\varphi_j \right) = I_{iz}\dot{\omega}_{iz} - \left( I_{ix} - I_{iy} \right)\omega_{ix}\omega_{iy} \quad (22)$$

where $m_i$ is the mass of the inner ring; $\ddot{x}_i$, $\ddot{y}_i$, $\ddot{z}_i$ are the displacement accelerations of the inner ring mass center; $I_{ix}$, $I_{iy}$, $I_{iz}$ are the moments of inertia of the inner ring; $\omega_{ix}$, $\omega_{iy}$, $\omega_{iz}$ are the angular velocities of the inner ring; $\dot{\omega}_{iy}$, $\dot{\omega}_{iz}$ are the angular accelerations of the inner ring; $F_x$, $F_y$, $F_z$, $M_y$, $M_z$ are the external loads and moments acting on the inner ring; and $r_{ij} = 0.5d_m - 0.5D_w f_i \cos\alpha_{ij}$, $f_i$ is the inner ring raceway curvature radius coefficient.

### 2.3. Power Loss Model of the Ball Bearing

The power loss of the ball bearing consists of ball-to-race interaction, ball-to-cage contact, cage-to-race contact, elastic hysteresis in rolling, and churning loss and drag loss in the cryogenic liquid.

(1)  The power loss of the ball sliding along the direction of the short axis:

$$H_M = \int_S v_b \mathrm{d}F \quad (23)$$

$$\mathrm{d}F = \mu(x,y)P(x,y)\mathrm{d}S \quad (24)$$

where $v_b$ represents the relative sliding velocity between two contact surfaces; $S$ is the area of the contact zone; and $\mathrm{d}F$ is the friction force on $\mathrm{d}S$. $\mu$ can be calculated from Equation (1).

(2)  Power loss due to the spinning sliding of the ball:

$$H_S = \int_{-a}^{a} \int_{-b[1-(x/a)^2]^{1/2}}^{b[1-(x/a)^2]^{1/2}} \mu(x,y)\omega_s \left( x^2 + y^2 \right)^{1/2} P_{\max}\left[ 1 - \left(\frac{x}{a}\right)^2 - \left(\frac{y}{b}\right)^2 \right]^{1/2} dydx \quad (25)$$

In Figure 10, $\omega_i$ is the angular velocity of the inner ring; $P_{\max}$ is the maximum contact stress between the ball and raceway; $a$ is the ellipse contact length; $b$ is the ellipse contact width; $\theta$ is the angle between $\mathrm{d}S$ and the long axis; and $l$ is the distance between $\mathrm{d}S$ and $o_e$.

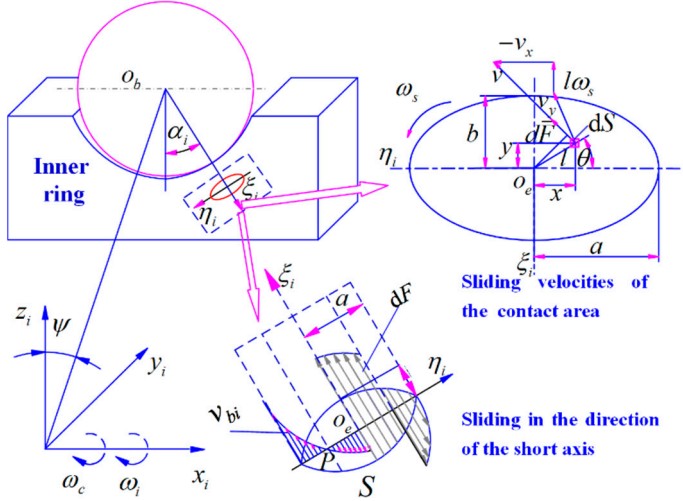

**Figure 10.** Sliding velocities of the area $\mathrm{d}S$ in the elliptical contact area.

(3)  Power loss due to the elastic hysteresis

$$M_E = \frac{3}{16} Qb\alpha_r \quad (26)$$

$$H_E = M_E\omega_c \quad (27)$$

where $Q$ is the contact load between the ball and inner/outer ring; and $\alpha_r$ is the elastic hysteresis coefficient and is equal to 0.07;

(4)　Power loss due to the contact between the ball/ ring and the cage:

$$H_{cage} = \mu_c Q_c v_{bc} + M_c \omega_c \tag{28}$$

where $\mu_c$ is the frictional coefficient between the cage and ball, whose expression is shown in Equation (1); and $v_{bc}$ is the sliding velocity between the ball and the cage pocket.

(5)　Power loss due to churning and drag

The drag force on the ball's surface is generally estimated by the empirical drag coefficient for a spherical body:

$$F_D = C_D \left[ \frac{1}{2} \rho V_b{}^2 A_{bf} \right] \tag{29}$$

$$H_{drag} = \frac{1}{2} F_D \omega_c D_w \tag{30}$$

where $F_D$ is the computed drag force; $C_D$ is the drag coefficient; $\rho$ is the effective density of the liquid; $V_b$ is the orbiting velocity of the ball, and $A_{bf}$ is the frontal area of the ball and liquid. The calculation expressions of these symbols refer to Ref. [33].

When a fluid passes through the cage, there will be a loss on the cylindrical surface, and the empirical formula for the churning moment on the surface is written as:

$$M_c = \frac{1}{2} f_c \rho U^2 A_{cf} r \tag{31}$$

$$H_c = M_c \omega_c \tag{32}$$

where $U$ is the mass average velocity of the fluid; $r$ is a reference radius from the center of rotation; $A_{cf}$ is the frontal area of the cage and liquid and $f_c$ is the friction factor. The calculation expressions of those symbols refer to Ref. [33].

(6)　Total power loss

The power loss of the contact areas is defined as:

$$C_{Total} = H_M + H_S + H_E + H_{cage} \tag{33}$$

The total power loss of the ball bearing is defined as:

$$H_{Total} = C_{Total} + H_{drag} + H_c \tag{34}$$

## 3. Analysis of Power Loss of the Ball Bearings of a Cryogenic Turbopump

The major parameters of ball bearings for cryogenic turbopumps are shown in Table 3. It is assumed that the ball bearing is dipped in LN2.

**Table 3.** Major parameters of ball bearings for cryogenic turbopumps.

| Item | Value |
| --- | --- |
| Bearing outside diameter (mm) | 218 |
| Bearing bore diameter (mm) | 118 |
| Bearing width (mm) | 40 |
| Ball diameter (mm) | 26.988 |
| Material of inner ring, outer ring, ball | 440C |
| Material of cage | PTFE |
| Material of raceway coating | Ag |

### 3.1. Component of Power Loss

When a ball bearing within a cryogenic turbopump start to work, the process can be divided into three stages, as shown in Figure 11. In the stage of (0~t3), the contact state between the ball and raceway is 440C-Ag coating, and then changes to 440C-PTFE transfer film in the stage of (t3~t4), before finally transferring to 440C-440C in the stage of (t4~t6) due to the severe wear of the contact interfaces. Due to the difference in contact interface, the values of the total power loss in the three stages are obviously different, the calculation results of which are shown in Figure 11 when the ball bearing is working at the given load and speed.

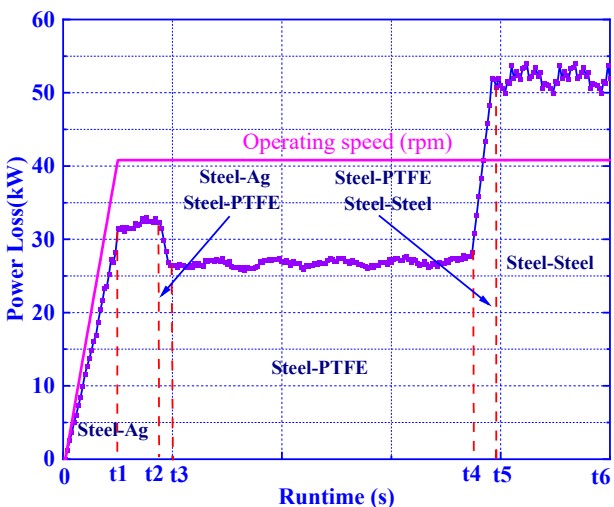

**Figure 11.** Power loss of ball bearing during the operation.

When the ball bearing is working at different speeds, the components of power loss in the three stages are shown in Figure 12.

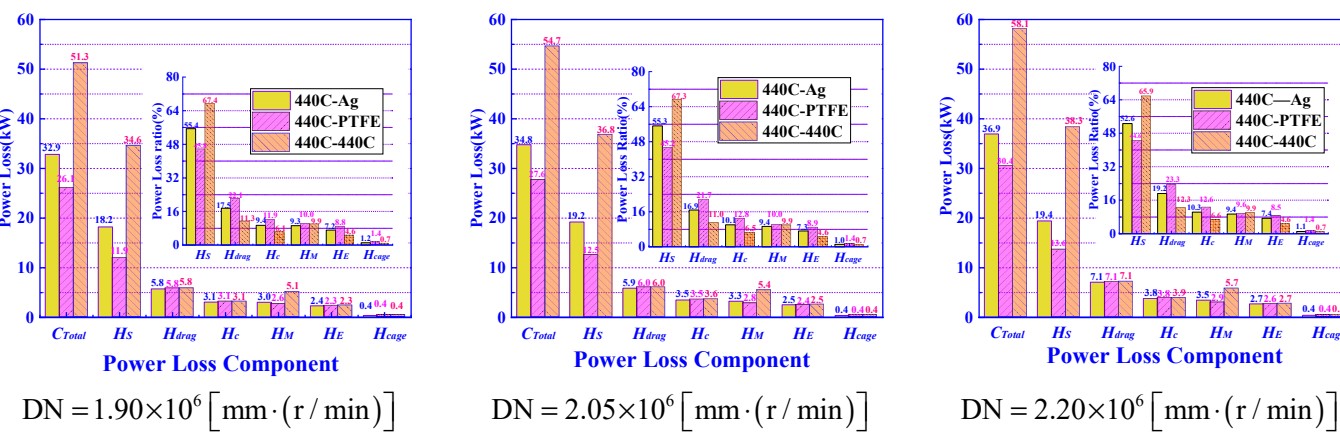

**Figure 12.** Power loss of the ball bearing under different DN values.

In Figure 12, it can be seen that: (1) The power loss of the ball bearing is the largest when the contact interface is in the stage of 440C-440C, the next is with 440C-Ag coating, the last is with 440C-PTFE transfer film. (2) $H_S$, $H_{drag}$ and $H_c$ represent more than 80% of the power loss of the ball bearing within a cryogenic turbopump, and, in particular, $H_S$ represents more than 45% of the loss, and will be discussed in detail in the next section. So, special attention should be paid to the spin slip of the ball for high-speed and high-load ball bearings in cryogenic turbopumps. (3) $H_S$ increases with the increase in speed, but the proportion of $H_S$ is dwindling, which is elaborated on in the next section. (4) The total of $H_{drag}$ and $H_c$ are increasing with DN due to the increasing churning speeds of the balls and

cage. (5) The contact interface has less effect on $H_{cage}$, $H_{drag}$, $H_E$ and $H_c$, but has a certain effect on $H_M$ due to the three contact interfaces with different frictional coefficients.

In addition, the power losses in the contact areas of the balls-inner ring and balls-outer ring are obviously different. In Figure 13, the image on the right shows a partial zoom-in of the ellipse area. It can be seen that the power loss in the contact area of the balls-inner ring is much larger than that in the contact area of the balls-outer ring, especially $H_S$, which receives special attention in the thermal field analysis described in the next section.

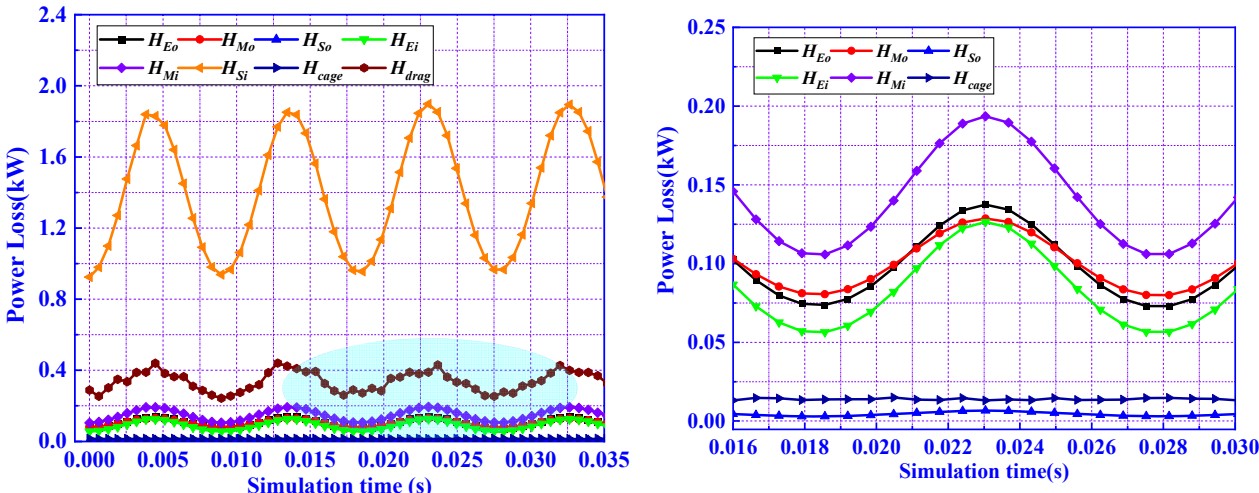

**Figure 13.** Power loss in the time domain.

### *3.2. Influence of Structural Parameters on Power Loss*

Because of the design constraints of ball bearings, the structure and dimensions of the cage and ball cannot be greatly changed, which has little effect on the $H_{cage}$, $H_c$ and $H_{drag}$ at constant load and speed. So, the major structural parameters, such as outer raceway diameter $D_e$, inner raceway diameter $d_i$, raceway curvature radius $f_i$ and $f_o$, etc. are selected as the influencing factors of the power loss in the contact areas of the balls-inner ring and balls-outer ring here. Moreover, in order to explain how the major structural parameters affect the power loss, the maximum value of $Pv_{i(o)}$, the spinning sliding velocity $v_{i(o)}$, the spinning angular velocity $\omega_{i(o)}$, the maximum contact stress $P_{i(o)}$, the spin-roll ratio $SR_{i(o)}$, the length of the elliptical contact area $a_{i(o)}$ and $b_{i(o)}$, etc. must be set. It should be noted that the results below are based on the assumption that the contact interfaces between balls and raceways are in the stage of 440C-Ag coating. The analysis results of 440C-PTFE transfer film and 440C-440C show the same trend and will not be discussed here.

#### 3.2.1. Influence of Outer Raceway Diameter $D_e$ on Power Loss

Outer raceway diameters $D_e$ in the range of 192.426 to 192.606 mm were studied, and the power loss, $\omega_{i(o)}$, $a_{i(o)}$, $b_{i(o)}$, $P_{i(o)}$ and other dynamic parameters are shown in Figure 14. It can be concluded that with the increase in $D_e$, the total power loss $C_{Total}$ in the contact area is increasing with the increase in $H_S$. $H_{So}$ is not plotted due to the small value, which can be reflected according to $Pv_o$. This is because the radial clearance of the ball bearing increases with the increase in $D_e$ and the contact angle $\alpha_{i(o)}$, which lead to a significant increase in $SR_{i(o)}$, $v_{i(o)}$ and $\omega_{i(o)}$, but a minor reduction in $P_{i(o)}$, $a_{i(o)}$ and $b_{i(o)}$. According to Equation (25), the increase in $H_{Si}$ is well-reasoned, as well as that of $Pv_i$. Therefore, minimizing the outer raceway diameter $D_e$ is one way to reduce the power loss if the design constraints of the bearing are satisfied.

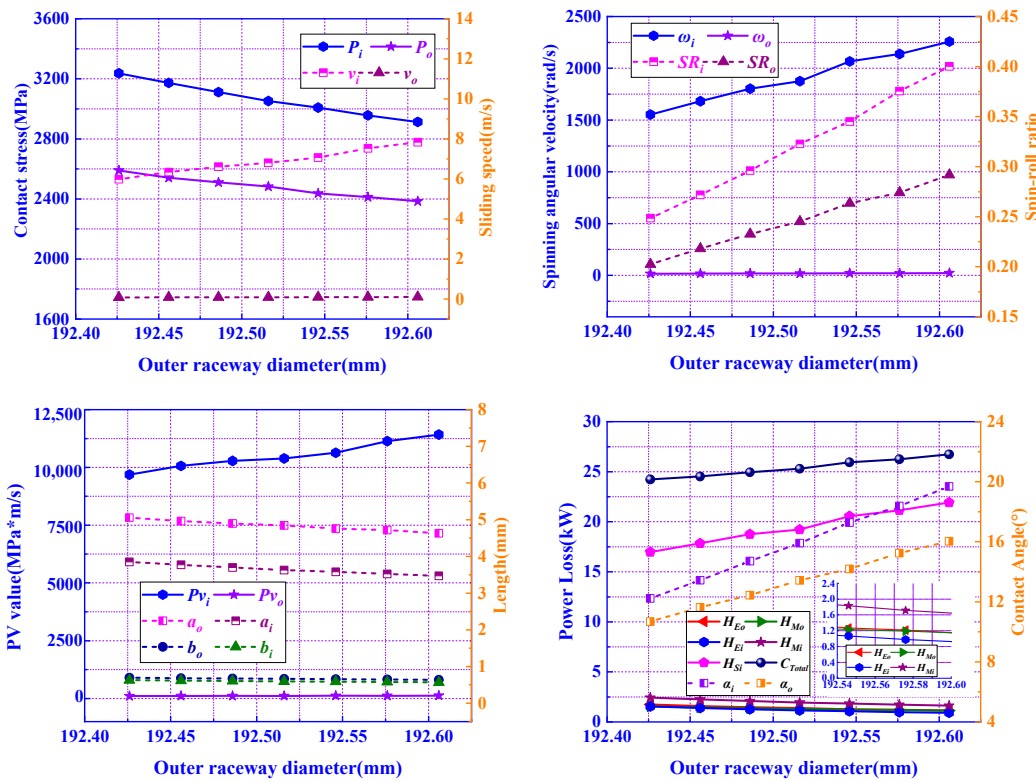

**Figure 14.** Influence of outer raceway diameter $D_e$ on the power loss.

### 3.2.2. Influence of Inner Raceway Diameter $d_i$ on Power Loss

Inner raceway diameters in the range of 138.368 to 138.548 mm were studied, and the power loss, $\omega_{i(o)}$, $a_{i(o)}$, $b_{i(o)}$, $P_{i(o)}$ and other dynamic parameters are shown in Figure 15. It can be concluded that with the increase in $d_i$, the total power loss $C_{Total}$ is decreasing due to the decrease in $H_{Si}$. $H_{So}$ is not plotted due to the small value, which can be reflected according to the $Pv_o$. This is because the radial clearance of the ball bearing decreased with the decrease in $d_i$, while the contact angle $\alpha_{i(o)}$ is also decreasing, which leads to a significant decrease in $SR_{i(o)}$, $v_{i(o)}$ and $\omega_{i(o)}$, but a minor increase in $P_{i(o)}$, $a_{i(o)}$ and $b_{i(o)}$. According to Equation (25), the decrease in $H_{Si}$ is well-reasoned, as well as that of $Pv_i$. Therefore, maximizing $d_i$ is one way to reduce the power consumption if the design constraints of the bearing are satisfied. Through the analysis of $D_e$ and $d_i$, a relatively small radial clearance and contact angle of the ball bearings for cryogenic turbopumps are suggested.

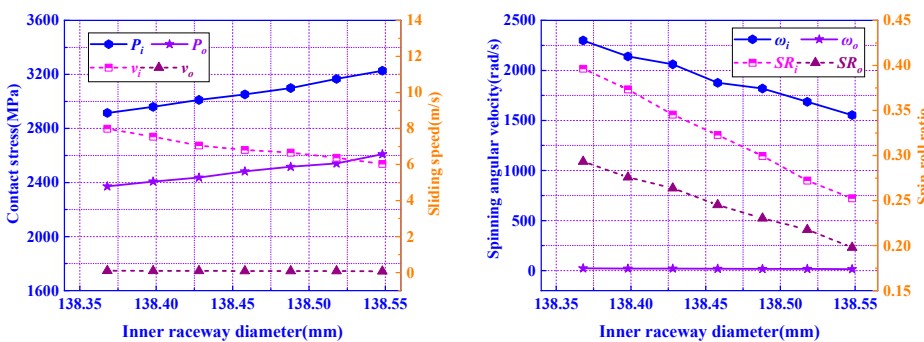

**Figure 15.** *Cont.*

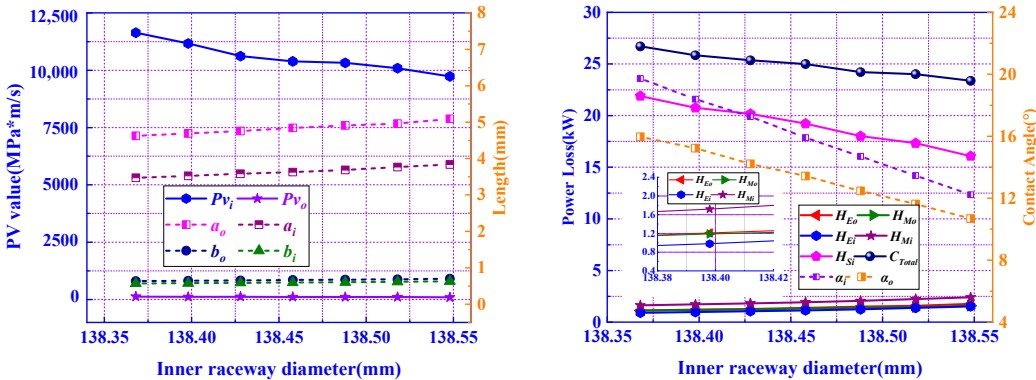

**Figure 15.** Influence of inner raceway diameter $d_i$ on power loss.

### 3.2.3. Influence of Inner Raceway Curvature Radius Coefficient $f_i$ on Power Loss

Inner raceway curvature radius coefficents in the range of 0.535 to 0.547 were studied, and the power loss, $\omega_{i(o)}$, $a_{i(o)}$, $b_{i(o)}$, $P_{i(o)}$ and other dynamic parameters are shown in Figure 16. It can be concluded that with the increase in $f_i$, the total power loss $C_{Total}$ is decreasing due to the decrease in $H_{Si}$. This is because, with the increase in $f_i$, $\alpha_i$ is decreasing, $P_i$ increases by about 7.75%, $v_i$ decreases by about 15.78%, so that $Pv_i$ and $H_{Si}$ are decreasing by about 3043 W. At the same time, the reduced amplitude of $H_{Mi}$ is about 237 W, and the increased amplitude of the other components is about 311 W, so $C_{Total}$ is decreasing gradually. However, when $f_i > 0.545$, the downward trend of $C_{Total}$ is not obvious, but the increase in $P_i$ is sustained, which will reduce the fatigue life of the bearing significantly. Therefore, there is a reasonable range of $f_i$ to balance the power loss and fatigue life/$P_i$.

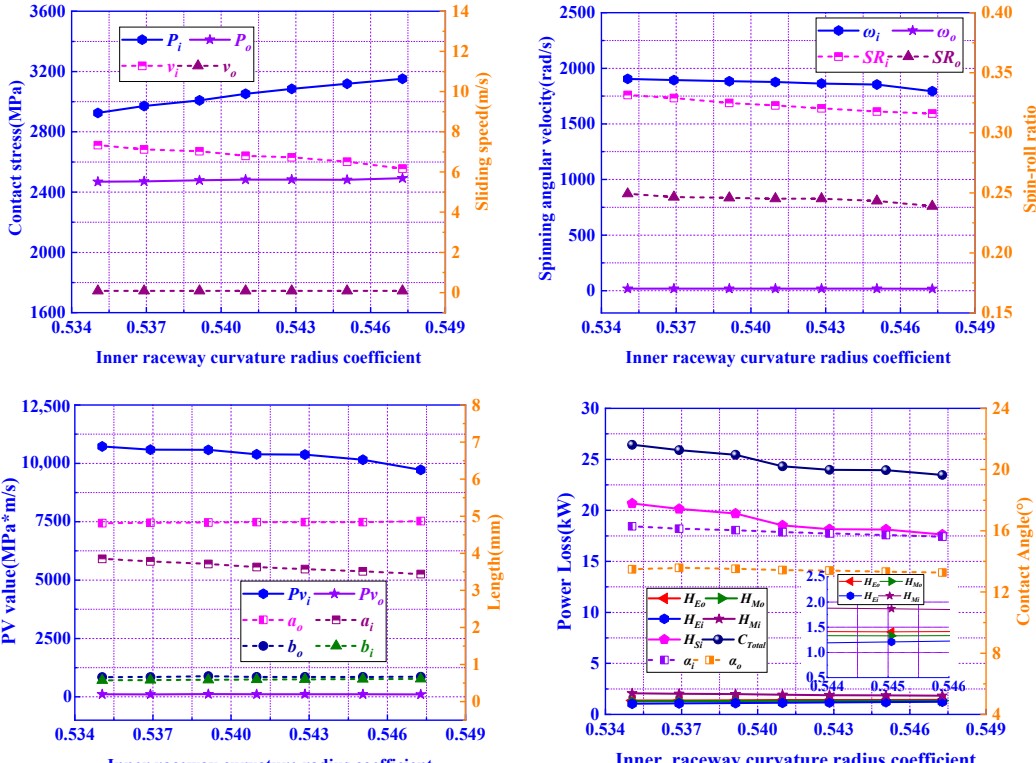

**Figure 16.** Influence of inner raceway curvature radius coefficient $f_i$ on power loss.

### 3.2.4. Influence of Outer Raceway Curvature Radius Coefficient $f_o$ on Power Loss

Outer raceway curvature radius coefficients in the range of 0.515 to 0.527 were studied, and the power loss, $\omega_{i(o)}$, $a_{i(o)}$, $b_{i(o)}$, $P_{i(o)}$ and other dynamic parameters are shown in Figure 17. It can be concluded that, with the increase in $f_o$, except for $P_o$ and $a_o$, the total power loss and the other performance expectations are not changed significantly. This can mainly be attributed to the fact that the power loss in the contact interfaces between the balls and the outer raceway is less than that in the contact interfaces between the balls and the inner raceway. It can be concluded that $f_o$ has a minor effect on the power loss compared to $f_i$, but a larger $f_o$ leads to a larger contact stress, which is harmful to the bearing's fatigue life.

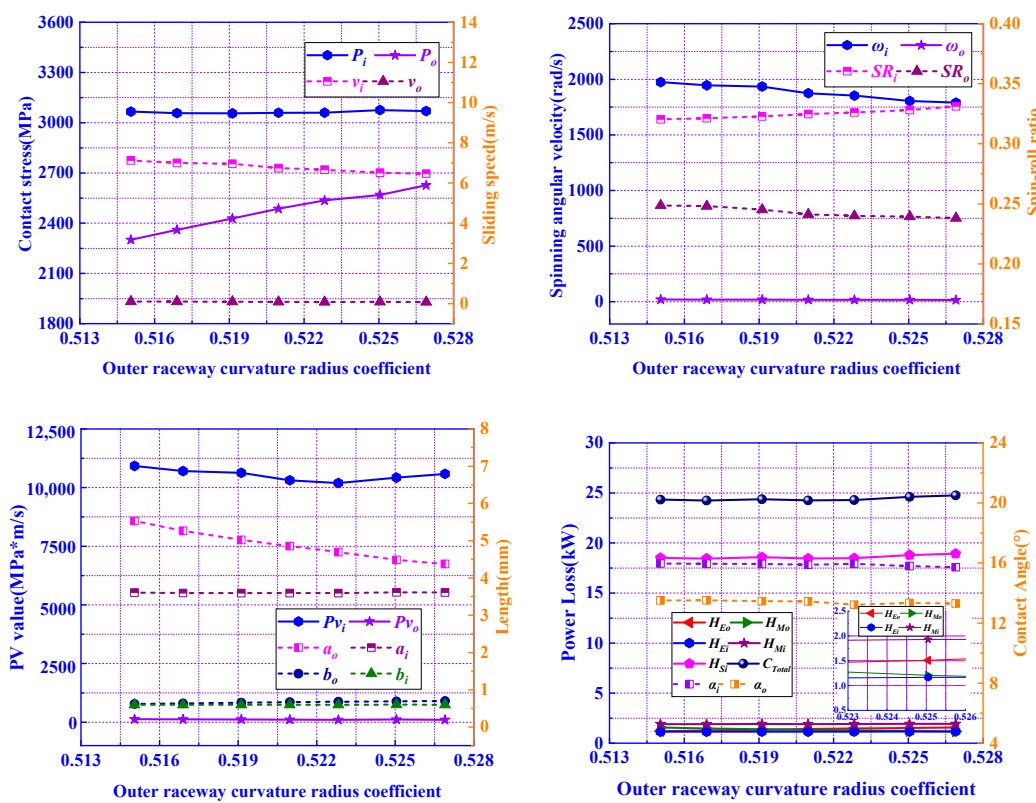

**Figure 17.** Influence of outer raceway curvature radius coefficient $f_o$ on power loss.

### 3.3. Influence of Working Conditions on Power Loss

Based on the analysis of the influence of structural parameters on the heat generation and power consumption of the bearing, those structural parameters were set to $D_e$ = 192.516 mm, $d_i$ = 138.458 mm, $f_i$ = 0.540, and $f_o$ = 0.520, and the influence of axial load $F_a$, radial load $F_r$ and bearing speed $n_i$ on the power loss are discussed.

### 3.3.1. Influence of Axial Load $F_a$ on Power Loss

The bearing speed $n_i$ was set to 17,000 rpm, the radial load $F_r$ was set to 20 kN, and the axial load $F_a$ was varied in the range of 28.0 to 40.0 kN. The results are shown in Figure 18. It can be seen that when the axial load $F_a$ is increased, the $P_{i(o)}$, $a_{i(o)}$, $\omega_{i(o)}$, $SR_{i(o)}$, $v_{i(o)}$, etc., are increased, which leads to $H_{Si}$ increasing rapidly, but $H_{So}$ experiences only a slight increase due to the minor amplitude of $v_o$. The other components of power loss all change only slightly.

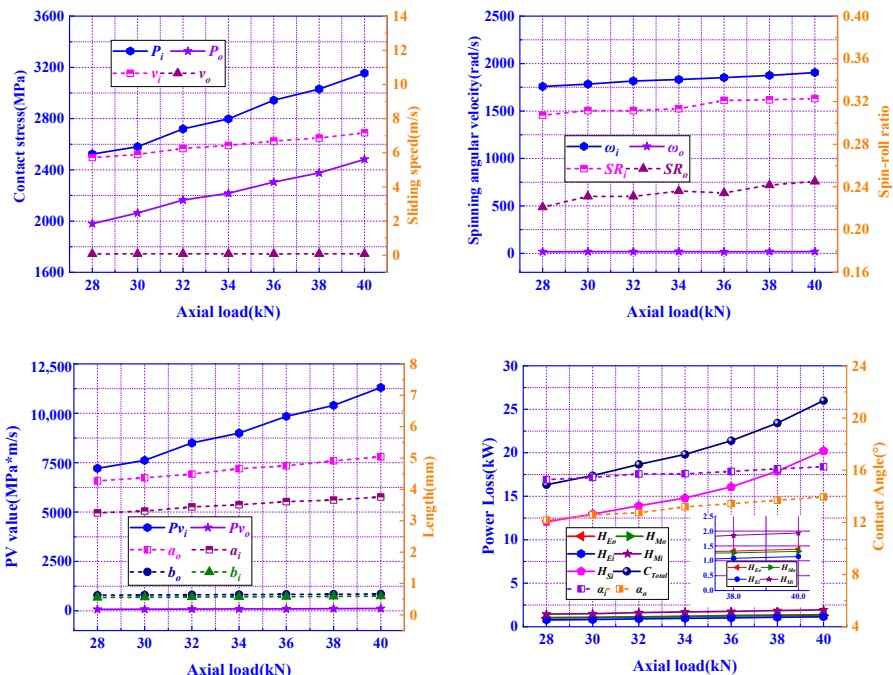

**Figure 18.** Influence of axial load $F_a$ on power loss.

### 3.3.2. Influence of Radial Load $F_r$ on Power Loss

The bearing speed $n_i$ was set to 17,000 rpm, the axial load $F_a$ was set to 40 kN, and the radial load $F_r$ was varied in the range of 8.0 to 20.0 kN. In Figure 19, when $F_r = 8$ kN, $C_{Total}$ is 24,325 W and when $F_r = 20$ kN, $C_{Total}$ is 25,997 W, the power loss increased by approximately 6.87%. It can be seen that the power loss increases slightly with increasing $F_r$. For a number of reasons, although the ratio of $F_a$ to $F_r$ is in the range of 2.0 to 5.0, the number of the loaded balls is not varied, which can be reflected by the slight change of, $P_{i(o)}$, $a_{i(o)}$, $\omega_{i(o)}$, $SR_{i(o)}$, $v_{i(o)}$, etc. Of course, a ball bearing with a reasonable radial clearance is the precondition.

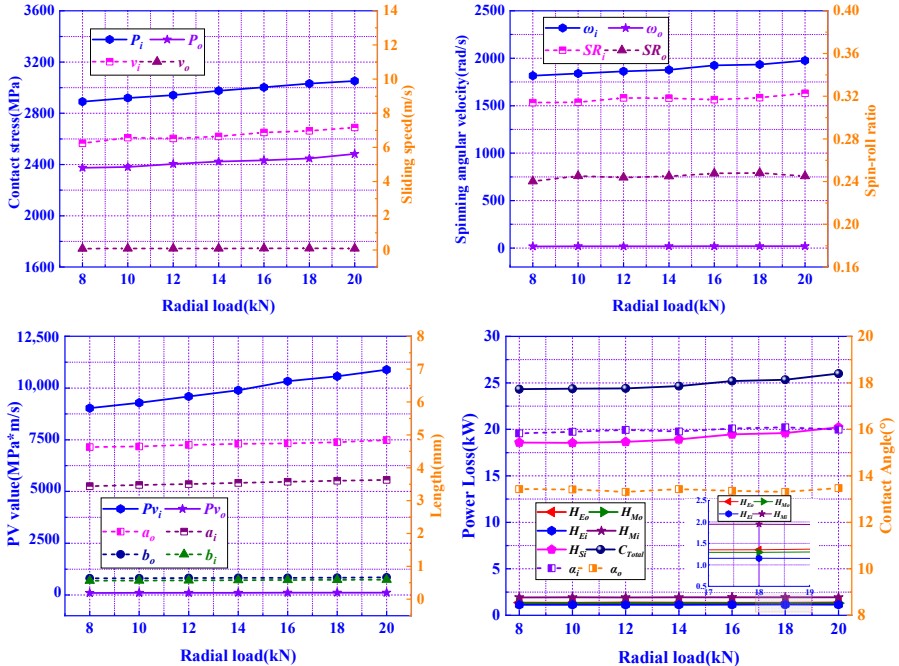

**Figure 19.** Influence of radial load $F_r$ on power loss.

### 3.3.3. Influence of Bearing Speed $n_i$ on Power Loss

The axial load $F_a$ was set to 40 kN, the radial load $F_r$ was set to 20 kN, and the bearing speed $n_i$ was varied in the range of 6000 to 18,000 rpm. The results are shown in Figure 20. It can be seen that the total power loss $H_{Total}$ is increasing with the increase in $n_i$, except for $H_{Si}$, the total of $H_{drag}$ and $H_c$ is larger and larger. For a number of reasons, due to the increase in bearing speed $n_i$, $\alpha_i$ trends to be larger and $\alpha_o$ trends to be smaller, which leads to a larger $SR_i$ and smaller $SR_o$. Although $P_i$ is decreasing with the increase in bearing speed $n_i$, $v_i$, $Pv_i$ and $H_{Si}$ are all increasing.

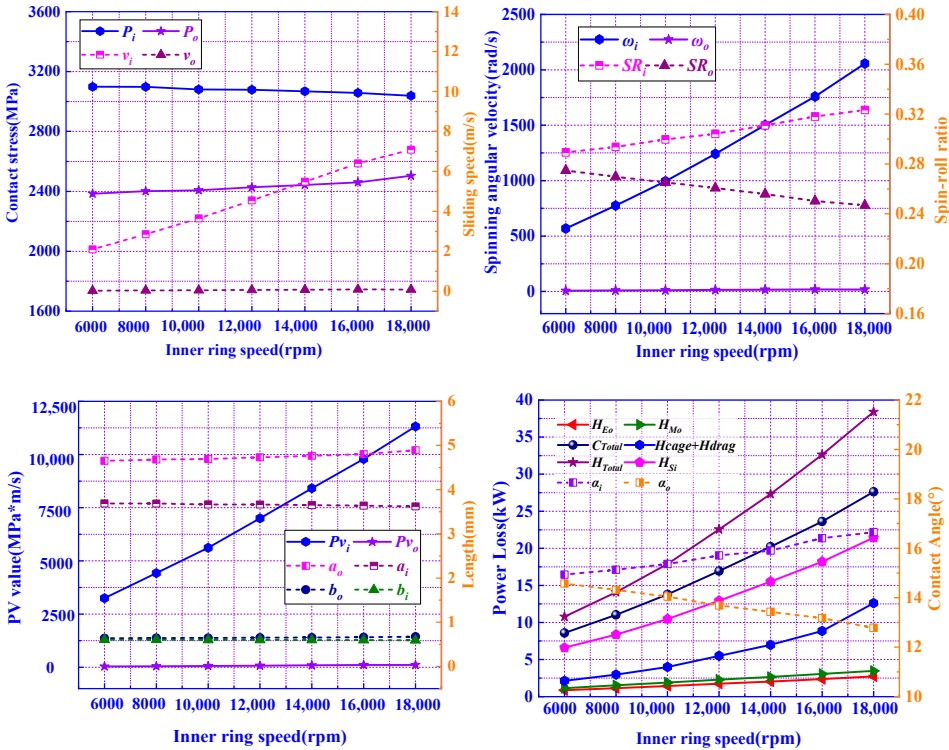

**Figure 20.** Influence of bearing speed $n_i$ on power loss.

## 4. Temperature Field Analysis and Test Verification

In order to verify the correctness of the theoretical models, a test rig was developed to test the ball bearings in a cryogenic turbopump [36,37], which uses the LN2 to build the cryogenic environment (Figure 21). The temperature of the outer ring is measured in real-time.

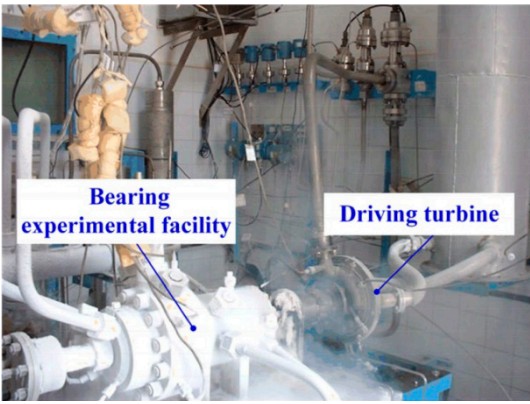

**Figure 21.** Ultra-low temperature ball bearing testing machine.

Based on the structure of the test rig, a coupled fluid-thermal finite element model was built in ANSYS Fluent to evaluate the temperature of the outer ring. In order to improve computational efficiency, only a part of the rotor system of the test rig is included in the model, as shown in Figure 22. In Figure 22, T2 and T4 are the tested ball bearings, and the meaning of the other symbols is explained in Table 4. The relative parameters of LN2 and the bearing materials are shown in Table 5.

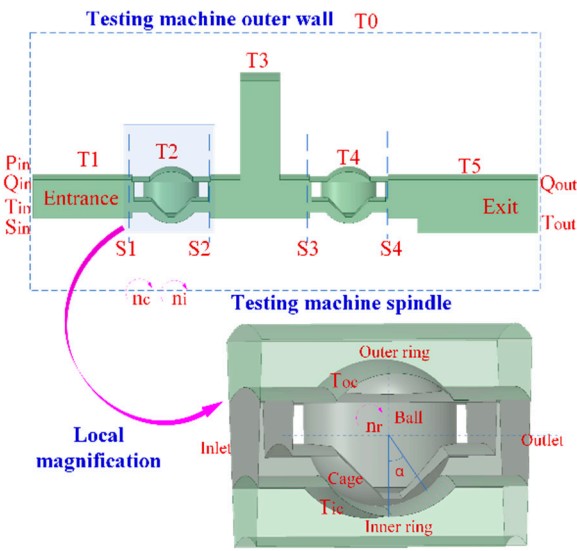

**Figure 22.** A coupled fluid-thermal finite element model.

**Table 4.** Flow velocity, pressure, and temperature boundary conditions.

| No. | Type | Variable |
|---|---|---|
| 1 | Inlet liquid nitrogen supply speed, pressure, temperature, area | $Q_{in}$, $P_{in}$, $T_{in}$, $S_{in}$ |
| 2 | Outlet pressure, temperature | $P_{out}$, $T_{out}$ |
| 3 | cage revolution speed, ball rotation speed | nc, nr |
| 4 | Inner surface temperature, outer surface temperature | $T_{ic}$, $T_{oc}$ |

**Table 5.** Relative parameters of LN2 and bearing materials.

| Medium | Density kg/m$^3$ | Specific Heat J/(kg·K) | Thermal Conductivity W/(m·K) | Viscosity kg/m-s | Moles kg/kmol |
|---|---|---|---|---|---|
| LN2 | 808.4 | 1040 | 0.026 | 0.0001 | 28.01 |
| PTFE | 2160 | 960 | 0.25 | - | - |
| 440C | 7750 | 481 | 29.3 | - | - |

The axial load $F_a$ was set to 40 kN, the radial load $F_r$ was set to 20 kN, and the bearing speeds $n_i$ were 16,000 and 17,000 rpm. The motion states of the balls and cage and the power loss of every contact interface were calculated using the theoretical models in Section 2.2.2, as shown in Table 6.

**Table 6.** Bearing motion boundary condition and power loss for one ball.

| $n_i$ (rpm) | $n_c$ (rpm) | $n_r$ (rpm) | $\alpha$ (°) | Ball-Outer Raceway (W) | Ball-Inner Raceway (W) | Ball-Cage (W) | Ball-Liquid (W) | Cage-Liquid (W) |
|---|---|---|---|---|---|---|---|---|
| 16,000 | 6805 | 43,177 | 17.6 | 190 | 1638 | 30 | 443 | 238 |
| 17,000 | 7239 | 45,833 | 18.0 | 211 | 1729 | 32 | 462 | 271 |



According to the test conditions, the temperature, pressure, and flow of LN2 are shown in Table 7.

**Table 7.** Temperature, pressure, and flow of LN2.

| $T_{in}$ (K) | $T_{ic}$ (K) | $T_{oc}$ (K) | $T_{out}$ (K) | $P_{in}$ (MPa) | $P_{out}$ (MPa) | $Q_{in}$ (kg/s) |
|---|---|---|---|---|---|---|
| 80 | 98 | 98 | 88 | 3.6 | 3.45 | 14 |

The results of the LN2 flow lines and temperature distribution are shown in Figures 23 and 24. It can be seen that the highest temperature is in the contact area between the ball and the inner raceway. The temperatures of the outer ring for T2 of the bearing under the different speeds are compared with the experimental results in Figure 25, as shown in Figure 26. The results show that the maximum error between the theoretical calculation result and the experimental result is less than 1.1%, which verifies the correctness of the dynamic and power loss models.

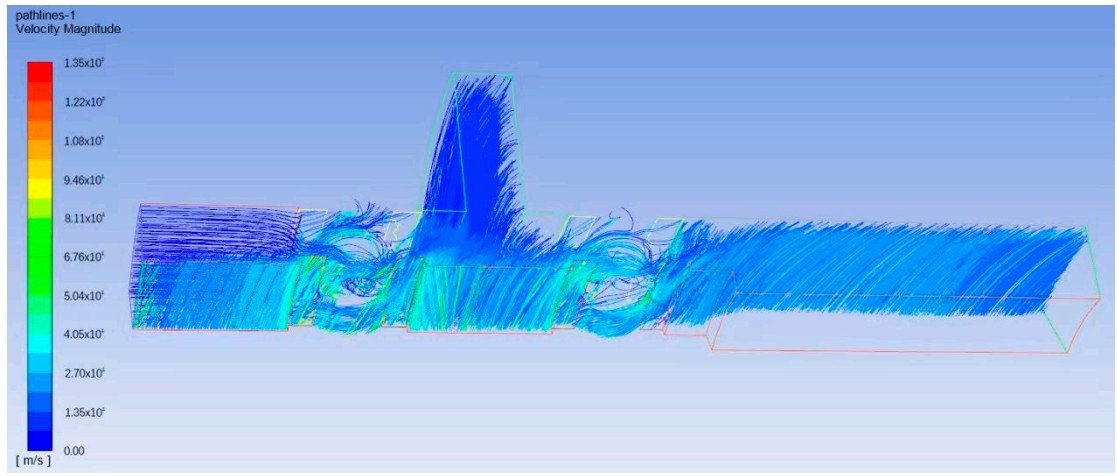

**Figure 23.** LN2 flow lines.

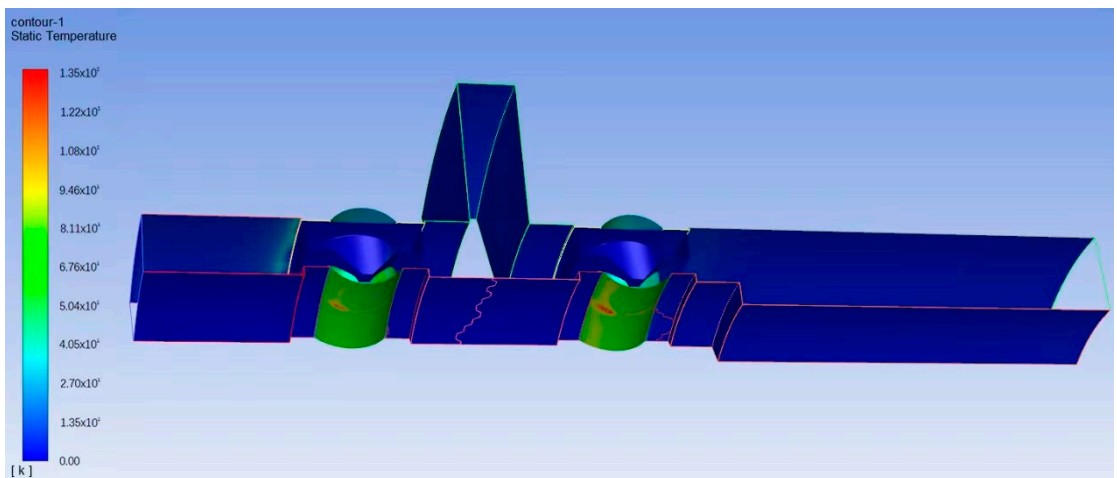

**Figure 24.** Temperature distribution.

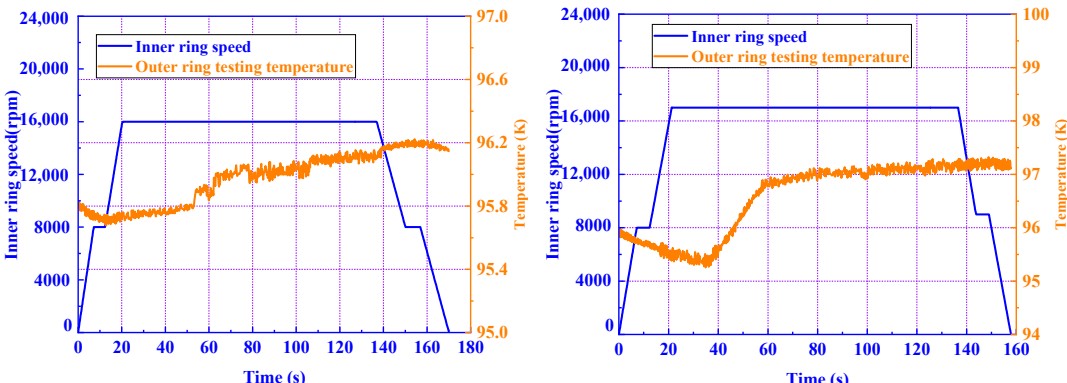

**Figure 25.** Experimental results for the temperature of the outer ring under 16,000 rpm and 17,000 rpm.

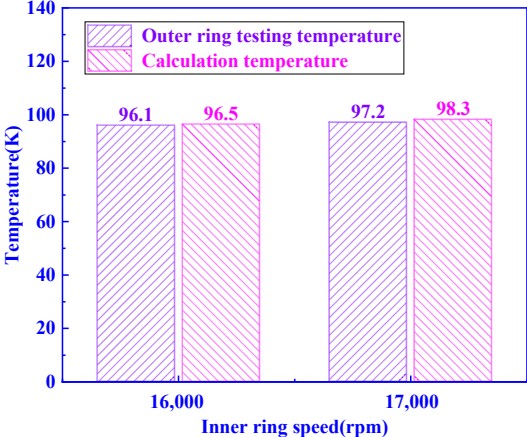

**Figure 26.** Comparison of experimental and theoretical results.

## 5. Conclusions

1. The total of $H_S$, $H_{drag}$, and $H_c$ represents more than 80% of the power loss of a ball bearing within a cryogenic turbopump, and in particular $H_{Si}$ represents the largest percentage (over 45%) throughout. So special attention should be paid to the spin-roll ratio $SR_i$ of the ball, which can be a key indicator for this type of ball bearing. At the same time, $H_{drag}$ and $H_c$ cannot be ignored when the ball bearing is working at high speed. The structural design of the cage and the flow of cryogenic fluid should be the focus of the next study.

2. A relatively small radial clearance and contact angle of a ball bearing within a cryogenic turbopump are suggested.

3. An inner raceway curvature radius coefficient $f_i$ with a larger value is suggested to reduce the power loss, but this will increase the maximum contact stress $P_i$ significantly. Therefore, there is a reasonable range of $f_i$ to balance power loss and fatigue life. The outer raceway curvature radius coefficient $f_o$ has a minor effect on the power loss compared to $f_i$, but a larger $f_o$ leads to a larger contact stress $P_o$ that is harmful to the bearing's fatigue life. Therefore, a relatively small value of $f_o$ is suggested. For the ball bearing in this paper, $f_i = 0.540$ and $f_o = 0.520$ are suggested.

4. When a ball bearing is working at a larger ratio of $F_a$ to $F_r$, the power loss of the ball bearing does not change much. A larger axial force $F_a$ is the key factor to impact the working states of the ball bearing, which leads to a significant change in the power loss.

**Author Contributions:** Conceptualization, W.Z. and S.D.; methodology, W.Z.; software, C.Z.; validation, C.Z.; X.M. and L.L.; formal analysis, W.Z.; investigation, W.Z.; resources, L.L.; data curation, X.M.; writing—original draft preparation, C.Z.; writing—review and editing, W.Z.; visualization,

C.Z.; supervision, S.D.; project administration, W.Z.; funding acquisition, W.Z. All authors have read and agreed to the published version of the manuscript.

**Funding:** This research was financially supported by the Youth Program of the National Natural Science Foundation of China (51905152).

**Institutional Review Board Statement:** Not applicable.

**Informed Consent Statement:** Not applicable.

**Data Availability Statement:** Not applicable.

**Conflicts of Interest:** The authors declare that they have no known competing financial interests or personal relationships that could have appeared to influence the work reported in this paper.

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
