# Peer review of "Research on the Power Loss of High-Speed and High-Load Ball Bearing for Cryogenic Turbopump"

_machines, doi:10.3390/machines10111080_

Round 1
Reviewer 1 Report
This is an interesting study and the authors have used unique approach for theoretical modelling. The paper is generally well written and structured. Few minor errors are uploaded.

Author Response
Dear reviewer and editor:
Thank you very much for giving us an opportunity to revise our manuscript. We appreciate the editor and reviewers for their positive and constructive comments and suggestions on our manuscript. We have studied the reviewer’s comments carefully and made revisions marked in red in the paper. We have tried our best to revise our manuscript according to the comments. We appreciate for Editors/Reviewers’ warm work earnestly and hope that the correction will meet with approval.
The main corrections in the paper and the responses to the reviewers’ comments are as follows:
(1)ball-disk can be replaced with ball-disc to maintain uniformity in the manuscript.
Response: Thanks for your suggestion! Ball-disk has been replaced with ball-disc.
(2)Figure 2 is a drawing from the book, so a reference to the book can be added to the figure title.
Response: Thanks for your suggestion! The revision has been done in Fig.2 title.
(3)“Mpa” need to be replaced with“MPa” in the manuscript.
Response: Thanks for your suggestion! “Mpa” has been replaced with“MPa” in the manuscript.
Reviewer 2 Report
Manuscript Number: Machines-2018146
Title: Research on the power loss of high-speed and high-load ball 2 bearing for cryogenic turbopump
Decision: Minor revision
Article Type: Article
This paper is aiming at the high-speed and high-load ball bearing for the cryogenic turbopump, the frictional coefficients of bearing’s contact surfaces are measured using the ball-on-disk experiments.. I think it should be reviewed and missing information should be added before it is published in the journal. The article is, in general, well written but there are some issues that article should consider to revise in order to improve its quality. Some comments were done in this way:
Ø In the abstract, it is necessary to prepare the reader to read the article. The abstract, according to the reviewer, is not a mini-paper but a quick tool to help readers decide whether they will read the rest of the paper. The rate of several remarkable improvements obtained as a result of the article should be given.
Ø The chemical composition of the material exposed to cryogenic heat is very important. The cryogenic temperature shows different metallurgical changes according to the Fe, Ni, Co, Mg, Mn ratios in the material. Therefore, the chemical composition of outer ring, inner ring, retainer and ball should be given.
Ø In Chapter 4 (4.Temperature field analysis and test verification) it is said “The temperature of outer ring is measured in real-time”. A graphic showing how the temperature is measured and the change of the outer ring temperature should be added.
Ø The times of the experiments carried out should be given in terms of time as well as the important test times given on rpm. Thus, it can be seen how long the materials exposed to -175°C are exposed to.
Ø The experimental setup was set up and the experiments were carried out, but mostly focused on theoretical knowledge. The effect of cryogenic treatment on bearing life should be examined. For this, the metallurgical effects of LN2 on the outer ring, inner ring, retainer and ball should be investigated and revealed.
Ø When the results are given, the improvement obtained as a result of the experiments should be given as a percentage.
After making the above corrections would recommend this article for publication in Machines.
Author Response
Dear reviewer and editor:
Thank you very much for giving us an opportunity to revise our manuscript. We appreciate the editor and reviewers very much for their positive and constructive comments and suggestions on our manuscript. We have studied the reviewer’s comments carefully and made revisions marked in red in the paper. We have tried our best to revise our manuscript according to the comments. We appreciate for Editors/Reviewers’ warm work earnestly and hope that the correction will meet with approval.
The main corrections in the paper and the responses to the reviewers’ comments are as follows:
(1)In the abstract, it is necessary to prepare the reader to read the article. The abstract, according to the reviewer, is not a mini-paper but a quick tool to help readers decide whether they will read the rest of the paper. The rate of several remarkable improvements obtained as a result of the article should be given.
Response: Thanks for your suggestion! Some remarkable results of the article have been added to the abstract.
(2)The chemical composition of the material exposed to cryogenic heat is very important. The cryogenic temperature shows different metallurgical changes according to the Fe, Ni, Co, Mg, Mn ratios in the material. Therefore, the chemical composition of outer ring, inner ring, retainer and ball should be given.
Response: Thanks for your suggestion! In this paper, I have given the material types for the outer ring, inner ring, retainer, and ball in Tab.3, which are the common materials for rolling bearings. The chemical composition of these materials refers to the relative standards, such as ASTM A276-2017.
Table 3. Major parameters of ball bearing.
Item |
Value |
Bearing outside diameter (mm) |
218 |
Bearing bore diameter (mm) |
118 |
Bearing width (mm) |
40 |
Ball diameter (mm) |
26.988 |
Material of inner ring, outer ring, ball |
440C |
Material of cage |
PTFE |
Material of raceway coating |
Ag |
(3)In Chapter 4 (4.Temperature field analysis and test verification) it is said “The temperature of outer ring is measured in real-time”. A graphic showing how the temperature is measured and the change of the outer ring temperature should be added.
Response: Thanks for your suggestion! The experimental results for the temperature of the outer ring under 16000rpm and 17000rpm have been added in figure.25.
Figure 25. Experimental results for the temperature of outer ring under 16000rpm and 17000rpm
(4)The times of the experiments carried out should be given in terms of time as well as the important test times given on rpm. Thus, it can be seen how long the materials exposed to -175°C are exposed to.
Response: Thanks for your suggestion! The time the material is exposed to -175°C is 200s. The details about the rig and the experimental process can refer to Ref.[29] from our team.
[29]Liu F, Su B, Zhang G, et al. Development of a Cryogenic Tester with Air Bearing to Test Sliding-Rolling Contact Friction[J]. Lubricants, 2022, 10(6): 119.
(5)The experimental setup was set up and the experiments were carried out, but mostly focused on theoretical knowledge. The effect of cryogenic treatment on bearing life should be examined. For this, the metallurgical effects of LN2 on the outer ring, inner ring, retainer and ball should be investigated and revealed.
Response: Thanks for your suggestion! In this paper, the power loss of the ball bearing is focused on. For rolling bearing, the cryogenic treatment with -50℃~-70℃ is used in the phase of heat treatment to reduce the retained austenite, and this is a benefit to the bearing’s precision. Whether LN2 has metallurgical effects on the outer ring, inner ring, retainer, and ball in a short time, is a new research field and has not been discussed in this paper.
(6)When the results are given, the improvement obtained as a result of the experiments should be given as a percentage.
Response: Thanks for your suggestion! Some results are given as a percentage.
